# Overexpression of Renin-B Induces Warburg-like Effects That Are Associated with Increased AKT/mTOR Signaling

**DOI:** 10.3390/cells11091459

**Published:** 2022-04-26

**Authors:** Janine Golchert, Doreen Staar, Jonathan Bennewitz, Miriam Hartmann, Nadin Hoffmann, Sabine Ameling, Uwe Völker, Jörg Peters, Heike Wanka

**Affiliations:** 1Institute of Physiology, University Medicine Greifswald, 17475 Greifswald, Germany; janine.golchert@med.uni-greifswald.de (J.G.); doreen.staar@med.uni-greifswald.de (D.S.); jonathanbennewitz@aol.de (J.B.); miriam.hartmann@med.uni-greifswald.de (M.H.); nadin.hoffmann@med.uni-greifswald.de (N.H.); heike.wanka@med.uni-greifswald.de (H.W.); 2Department of Functional Genomics, Interfaculty Institute for Genetics and Functional Genomics, University Medicine Greifswald, 17475 Greifswald, Germany; sabine.ameling@uni-greifswald.de (S.A.); voelker@uni-greifswald.de (U.V.); 3Partner Site Greifswald, DZHK (German Center for Cardiovascular Research), 17475 Greifswald, Germany

**Keywords:** H9c2 cells, cytosolic renin, transcriptome analysis, Warburg effect, AKT/mTOR signaling

## Abstract

The classical secretory renin-a is known to be involved in angiotensin generation, thereby regulating not only blood pressure, but also promoting oxidative stress as well as apoptotic and necrotic cell death. In contrast, another cytosolic renin isoform named renin-b has been described, exerting protective effects under ischemia-related conditions in H9c2 cardiomyoblasts. Using microarray-based transcriptome analyses, we aimed to identify the signaling pathways involved in mediating cardioprotection in H9c2 cells overexpressing renin-b. By transcriptome profiling, we identified increased gene expression of several genes encoding glycolytic enzymes and glucose transporters, while the transcript levels of TCA-cycle enzymes were decreased. Complementing data from metabolic analyses revealed enhanced glucose consumption and lactate accumulation due to renin-b overexpression. Renin-b overexpression further stimulated AKT/mTOR signaling, where numerous genes involved in this pathway showed altered transcript levels. For AKT, we also detected enhanced phosphorylation levels by means of Western blotting, suggesting an activation of this kinase. Moreover, analysis of the ROS levels identified an increase in ROS accumulation in renin-b-overexpressing cells. Altogether, our data demonstrate that renin-b overexpression induces the metabolic remodeling of H9c2 cells similar to that seen under oxygen deprivation. This metabolic phenotype exerting so-called aerobic glycolysis is also known as the Warburg effect.

## 1. Introduction

Altered cellular energy metabolism, known as metabolic remodeling, plays a central role in cancer cells, proliferating cells, as well as in the ischemic heart. However, it often remains unclear whether these molecular alterations are adaptive or maladaptive responses. Therefore, the identification of the key pathways and endogenous compounds involved in metabolic remodeling are relevant aspects of research.

Catabolic glucose metabolism encompasses glycolysis, the pentose-phosphate pathway (PPP), and oxidative phosphorylation (OXPHOS) to generate adenosine triphosphate (ATP) and the reducing equivalent nicotinamide adenine dinucleotide phosphate (NADPH). Glucose transporters import glucose into the cell, which is metabolized by a series of glycolytic enzymes to generate pyruvate. A flux of glucose-6-phosphate into the PPP provides NADPH, glutathione, and the nucleotide precursor ribose-5-phosphate [1]. NADPH and glutathione are essential factors involved in protection against cellular damage caused by the accumulation of reactive oxygen species (ROS), especially during anabolic reactions. Pyruvate is further metabolized by the tricarboxylic acid (TCA) cycle delivering the electron carriers nicotinamide adenine dinucleotide (NAD) + hydrogen (H) (NADH), and the hydroquinone form of flavin adenine dinucleotide (FADH_2_) to mitochondrial electron transport chain complexes I, II, and III. Thus, a proton gradient across the inner mitochondrial membrane is generated, which, in turn, is necessary for ATP generation in complex V of the respiratory chain [2]. Cells adapt by modulating these metabolic pathways in response to intracellular energy demands and extracellular stimuli, as well as in response to oxidative changes.

Former studies by our group showed that, in rats, a non-secretory renin isoform, termed renin-b, is upregulated during myocardial infarction in vivo [3], as well as in rat primary cardiomyocytes or cardiomyoblasts in vitro under ischemia-related conditions, such as glucose deprivation and hypoxia [4,5,6]. In rats, renin-b is encoded by the alternative renin transcript *Ren(1a–9)* and is generated by translation initiation at start codon AUG located in exon 2, hence lacking the classical exon 1. However, exon 1 contains the signal sequence for co-translational transport to the rough endoplasmic reticulum. Therefore, the translation of renin-b occurs at free ribosomes, and renin-b is located within the cytosol and is partially imported into the mitochondria [7,8,9]. Transcripts lacking exon 1 have also been described in mice [10,11] and humans [12]. These alternative transcripts contain additional 5′ untranslated regions (5′UTR) upstream of exon 2, which may have unknown regulatory functions. In contrast to cells overexpressing the classical secretory renin-a, H9c2 cells overexpressing renin-b (with [*Ren(1a–9)*] or without [*Ren(2–9)*] its 5′UTR) were protected against starvation conditions, such as oxygen and glucose deprivation. Although these cells exhibited increased basal apoptosis rates, starvation-induced increases in the rates of necrosis, apoptosis, and reactive oxygen species (ROS) generation, as well as the starvation-induced decrease in the ATP levels, were partially or completely abolished [4,6,9,13,14]. Furthermore, renin-b [*Ren(2–9)*] overexpression leads to a Warburg-like phenotype characterized by increased glucose consumption and increased extracellular lactate accumulation [15]. The Warburg effect comprises the metabolic shift from OXPHOS to more glycolysis, even under aerobic conditions [16]. Although ATP generation is less efficient, the Warburg effect provides a selective advantage by balancing ATP production and anabolic processes, such as protein or lipid biosynthesis, as well as by increasing cellular survival in the presence of fluctuating oxygen levels [17,18]. Key players in orchestrating these adaptations are adenosine monophosphate (AMP)-activated protein kinase (AMPK), protein kinase B (AKT), mitogen-activated protein kinase (MAPK), as well as the mechanistic target of rapamycin (mTOR) signaling [19].

To identify potential signaling cascades and microRNAs (miRNAs) involved in their alterations induced by renin-b, we performed transcriptome analyses using H9c2 cells overexpressing renin-b transcripts with or without its 5′UTR, respectively. In the present study, we demonstrate that renin-b overexpression in H9c2 cardiomyoblasts activates glycolytic metabolism, whereas the TCA cycle is suppressed, and that it is associated with an activation of AKT/mTOR signaling in vitro.

## 2. Materials and Methods

### 2.1. Cell Culture and Overexpression of Renin Transcripts

Rat cardiomyoblasts (H9c2) obtained from ATCC (Manassas, VA, USA) were cultured in DMEM (PAN-Biotech, Darmstadt, Germany) supplemented with 10% fetal bovine serum (FBS, PAN-Biotech), 100 U/mL penicillin, and 100 mg/mL streptomycin (PAN-Biotech) in tissue culture flasks at 37 °C in a humidified atmosphere of 5% CO_2_. H9c2 cells were transfected with empty pIRES vector or with pIRES vector containing the coding DNA sequence of renin-b consisting of exon 1a to exon 9 or exon 2 to exon 9 of the rat renin gene. The detailed structure of both renin transcript variants is given in Figure 1. Here, the transfection of H9c2 cells was carried out with minor modifications according to the polyethyleneimine (PEI) method reported by Boussif et al. [20]. Briefly, H9c2 cells were seeded at 2 × 10^5^ cells in 6-well plates 3 days prior to plasmid transfection. Immediately before transfection, the cells were replenished with fresh serum- and antibiotic-free culture medium. One microgram of plasmid DNA and 4 µL of PEI solution (Sigma-Aldrich, Taufkirchen, Germany) were diluted each in 100 µL OptiMEM solution (Gibco, Thermo Fisher Scientific, Waltham, MA, USA). The two solutions were mixed and incubated for 20 min at room temperature before carefully adding them to the cells. The cells were incubated for 6 h in serum-free medium followed by incubation for 18 h in fresh culture medium containing 10% FBS. To generate stable renin-b-overexpressing cell lines, transfected cells were cultured in the presence of G418 sulfate (430 µg/mL, Gibco) to ensure the survival of transfected cells only.

### 2.2. RT-qPCR

The total RNA from cells was isolated using peqGOLD TriFast™ (peqlab, VWR International GmbH, Darmstadt, Germany) according to the manufacturer’s protocol. RNA samples were purified using the RNA Clean-Up and Concentration Micro Kit (Norgen Biotek Corp., Thorold, Canada), the RNA concentration was determined using spectrophotometry (NanoDrop 8000, Thermo Fisher Scientific), and quality control was performed using an Agilent 2100 Bioanalyzer (Agilent Technologies, Santa Clara, CA, USA).

The reverse transcription of 1 µg of total RNA into cDNA was performed using a High Capacity cDNA Kit (Life Technologies, Thermo Fisher Scientific). For qPCR, 20 ng cDNA per sample (*n* = 3–4) was analyzed in duplicate using either the Rotor-Gene SYBR Green PCR Kit (Qiagen, Hilden, Germany) or Biozym Blue S’Green qPCR Kit Separate Rox (Biozym Scientific, Hessisch Oldendorf, Germany) and optimized primer pairs for the different transcripts, as well as the housekeeping gene tyrosine 3-monooxygenase/tryptophan 5-monooxygenase activation protein, zeta (*Ywhaz*) (Table 1). qPCR was performed using a Rotor-Gene Q (Qiagen). Data were analyzed using the threshold cycle number (Ct) in combination with the 2^−ΔΔCt^ method with *Ywhaz* as the housekeeping gene.

### 2.3. Transcriptome Analysis

The total RNA from cells was isolated using peqGOLD TriFast™ according to the manufacturer’s protocol. RNA samples were purified using the RNA Clean-Up and Concentration Micro Kit, RNA concentration was determined using spectrophotometry (NanoDrop 8000), and quality control was performed using an Agilent 2100 Bioanalyzer. Microarray analysis was carried out using individual RNA samples (*n* = 4), which were processed following the manufacturer’s instructions of the GeneChip^TM^ WT PLUS Reagent Kit (Thermo Fisher Scientific) and hybridized with GeneChip^TM^ Clariom D Rat Arrays (Thermo Fisher Scientific). Quality control of the hybridizations and data analysis were performed using Transcriptome Analysis Console (Thermo Fisher Scientific). The data were normalized using the Robust Multi-chip Analysis (RMA) algorithm.

To identify significantly differentially expressed genes (*q*–value < 0.05, fold change ≥ 1.3-fold), one-way ANOVA (with Bayes estimation) was performed, where *p*-values were corrected for multiple testing using Benjamini Hochberg adjustment. All data from transcriptome analyses are provided in Appendix A. Significantly differentially expressed genes were further subjected to in silico pathway analysis using the Ingenuity Pathway Analysis software (Ingenuity Systems, Inc. Redwood City, CA, USA). This enrichment analysis was based on all annotated rat genes in the database.

### 2.4. Glucose Consumption and Lactate Accumulation

An amount of 0.2 × 10^6^ pIRES or renin-b-overexpressing H9c2 cells were seeded into 6-well plates with complete DMEM medium to allow attachment at 37 °C in a humidified atmosphere of 5% CO_2_ for 3 days. The culture medium was replaced with 2 mL of fresh medium. Twenty-four hours later, the culture medium was collected for the determination of the glucose and lactate concentrations using the Biosen C line GF+ analyzer (EKF Diagnostics, Barleben, Germany) as described previously [15]. Medium without cells served as a control for calculating glucose uptake and lactate accumulation. The ratio of glucose uptake to extracellular lactate accumulation served to estimate the manner of glucose metabolism.

### 2.5. ROS Detection

For analyzing ROS, 1 × 10^5^ trypsinated cells were collected and resuspended in 500 µL of DMEM supplemented with 5 µL of MitoSOX Red mitochondrial superoxide indicator (5 µmol/L), 10 µL of dihydrorhodamine 123 (DHR, 5 µmol/L), or 10 µL of dihydroethidium (DHE, 2 µmol/L) (each Invitrogen, Thermo Fisher Scientific) for 30 min at 37 °C. Unstained cells were used as negative controls. After labeling, the cells were washed and resuspended in FACS buffer (Thermo Fisher Scientific). Fluorescence intensity was measured using a BD FACS Calibur system (Becton, Dickinson and Company, Franklin Lakes, NJ, USA) for analysis as described previously [13].

### 2.6. Western Blotting

For Western blot analyses, protein samples were prepared and analyzed as described before [4]. Briefly, proteins were isolated using RIPA buffer, and 25 µg per sample was separated by SDS-PAGE using 4–15% Criterion TGX gradient gels (BioRad Laboratories, Munich, Germany). Proteins were transferred to a nitrocellulose membrane by wet blotting and protein imaging was performed using UV transillumination on a Chemidoc XRS (Bio-Rad Laboratories). For the detection of target protein-specific primary antibodies [mTOR (7C10) Rabbit mAb, # 2983, 1:1500; Phospho-mTOR (Ser2448) (D9C2) XP Rabbit mAb, # 5536, 1:2000; AKT (pan) (C67E7) Rabbit mAb, # 4691, 1:10,000; Phospho-AKT (Ser473) (D9E) XP Rabbit mAb, # 4060, 1:5000, each Cell Signaling Technology, Danvers, MA, USA] and a horseradish peroxidase-conjugated secondary antibody (anti-rabbit IgG, HRP-linked Antibody, Cell Signaling, #7074, 1:5000) were used. The relative protein abundance was visualized by enhanced chemiluminescence detection using an image-capture system (Chemidoc XRS, BioRad Laboratories). For normalization, the total protein loading was detected by the UV transillumination of membranes.

### 2.7. Statistical Analyses

Data from independent experiments were normally distributed and analyzed by one-way ANOVA followed by Tukey’s test using GraphPad Prism (Graph Pad Software version 9.2.0, La Jolla, CA, USA). Single *p*–values < 0.05 vs. pIRES controls or as indicated are given in the figures. For the statistical analysis of the transcriptome data, see 2.3.

## 3. Results

### 3.1. Renin-B Overexpression Affects the Expression of Genes and microRNAs Known to Be Involved in a Warburg-like Phenotype

One hallmark of the Warburg effect is the increase in glycolysis while mitochondrial functions are suppressed. To elucidate a putative Warburg effect in renin-b-overexpressing cell lines, we used transcriptome profiling to examine alterations in the levels of transcripts encoding glycolytic enzymes, as well as enzymes of the TCA cycle. By RT-qPCR, the overexpression of the *Ren(1a–9)* and *Ren(2–9)* transcript levels was measured, resulting in 2.06- and 1.74-fold overexpression of Ren(1a–9) and Ren(2–9), respectively.

A heat map of significantly differentially expressed genes (*q* < 0.05, fold change ≥ |1.3|) involved in glucose metabolism and the TCA cycle is shown in Figure 2. There was a great overlap of differentially expressed genes between Ren(1a–9) and Ren(2–9) cells. In comparison to pIRES control cells, renin-b-overexpressing cell lines (Ren(1a–9) and Ren(2–9)) representing transcripts with or without 5′UTR, respectively, exhibited a marked increase in the transcript levels of genes encoding various glycolytic enzymes, while the transcript levels of several TCA cycle-related enzymes were reduced.

Among glycolysis-related enzymes, we found that the transcript levels for phosphofructokinase (*Pfkm)*, phosphoglycerate kinase 1 (*Pgk1*), muscle-specific phosphoglycerate mutase 2 (*Pgam2*), and 6-phosphofructo-2-kinase/fructose-2,6-bisphosphatase 3 (*Pfkfb3*) were increased in both cell lines, while hexokinase 1 encoding transcript *Hk1* was increased only in Ren(1a–9)-overexpressing cells. The final product of glycolysis, pyruvate, was catalyzed by the pyruvate dehydrogenase complex (PDH) to acetyl-CoA or by the pyruvate carboxylase (PC) to oxaloacetate after uptake into the mitochondria via mitochondrial pyruvate carriers 1 and 2 (MPC1/2). Neither the transcript levels of *Mpc1*/*Mpc2* nor *Pdh* subunits coding genes were differentially expressed in renin-b-overexpressing cells. In contrast, the expression of *Pdk2* coding for pyruvate dehydrogenase kinase 2, which inhibits the activity of PDH, was markedly increased in the Ren(1a–9) and Ren(2–9) cell lines. The expression of *Pc* was increased in Ren(1a–9) and increased in trend in Ren(2–9) cells.

As mentioned above, for genes encoding TCA cycle enzymes, we detected decreased mRNA levels of isocitrate dehydrogenase 3 alpha (*Idh3a*) and 3 beta (*Idh3b*), succinate CoA ligase ADP-forming (*Sucla2*) and GDP-forming beta subunits (*Suclg2*), as well as succinate dehydrogenase complex subunit D (*Sdhd*). In contrast, the transcript levels of oxoglutarate (α-ketoglutarate) dehydrogenase (*Ogdh*) were increased in Ren(1a–9) and Ren(2–9) cells. Moreover, the transcript levels of fumarate hydratase (*Fh*) were decreased exclusively in Ren(2–9) cells, while those of malate dehydrogenase 2 (*Mdh2*) were increased in Ren(1a–9) cells only.

Additionally, we found increased levels of several genes coding for members of the mitochondrial solute carrier family 25 (*Slc25a*) being involved in the transport of various TCA cycle metabolites across the inner mitochondrial membrane, such as dicarboxylates. Here, we show an increased expression of malate and succinate carrier-coding gene *Slc25a10* as well as oxoglutarate/malate carrier-coding gene *Slc25a11* in Ren(1a–9) and Ren(2–9) cells. The expression of the citrate transporter-coding gene *Slc25a1* was increased in Ren(1a–9) cells, but remained unchanged in Ren(2–9) cells.

Among the numerous known miRNAs involved in the regulation of Warburg-relevant target genes, we found decreased expression of miR-1291 in Ren(1a–9)-overexpressing cells (Figure 3) influencing the protein abundance of glucose transporter 4 (GLUT4). In contrast, the levels of AKT-regulating miRNAs miR-221 and miR-21 were increased, as well as the levels of miRNA let-7f-1 (Figure 3), which is known to influence the expression of *Mtor*.

### 3.2. Renin-B Overexpression Induces Metabolic Alterations Involved in a Warburg-like Phenotype

To further validate the hypothesis of an existing Warburg effect caused by renin-b overexpression, we tested whether these cells show altered glucose uptake and extracellular lactate accumulation. We also assayed if the cells exhibited an altered glucose-to-lactate ratio, indicating altered conversion of glucose to lactate. Additionally, we analyzed the transcript levels of genes encoding transporters necessary for the cellular uptake of glucose and influx/efflux of pyruvate.

Indeed, we observed significantly increased glucose uptake (Figure 4A) together with significantly enhanced extracellular lactate accumulation (Figure 4B) in both renin-b-overexpressing cell lines compared to pIRES control cells (Figure 4B). The ratio indicating the level of lactate production from glucose decreased significantly in Ren(2–9) cells only (Figure 4C). These results were supported by the finding of increased transcript levels of genes encoding glucose transporters (solute carrier family 2 (*Slc2a*) members), which are associated with increased cellular glucose uptake (Figure 4D). In Ren(1a–9) and Ren(2–9) cells, the transcript levels of both *Slc2a1* (GLUT1) and *Slc2a4* (GLUT4) were increased. In contrast, the mRNA levels of *Slc2a3* (GLUT3) were decreased in Ren(1a–9) and Ren(2–9) cells. However, despite the increased extracellular lactate accumulation, neither the gene expression of lactate dehydrogenase A (LDHA) catalyzing the conversion of pyruvate to lactate nor the gene expression of monocarboxylate transporters SLC16A1, 3, 7, and 8 (MCT1, 4, 2, and 3) involved in the influx/efflux of lactate were altered (data not shown).

In summary, renin-b overexpression was not only associated with increased expression of genes coding for glucose transporters and glycolytic enzymes, but also with increased uptake of glucose and accumulation of extracellular lactate. These findings support the hypothesis that renin-b may be involved in metabolic alterations, which is consistent with the manifestation of a Warburg-like phenotype.

### 3.3. Renin-B Increases AKT/mTOR Signaling as Detected by Transcriptome Profiling

Several signaling pathways, interplaying with enzymes and kinases involved in glucose metabolism, participate in the switch from oxidative phosphorylation (OXPHOS) to aerobic glycolysis. One of them is the phosphoinositide 3-kinase (PI3K) signaling pathway regulating glucose uptake and glycolysis via AKT and mTOR. By using pathway enrichment analyses (IPA), we identified 109 differentially expressed genes (*q* < 0.05, fold change ≥ |1.3|) coding for proteins involved in PI3K/AKT and mTOR signaling in Ren(1a–9) and/or Ren(2–9)-overexpressing cells as compared with pIRES controls. A heat map of these genes is shown in Figure 5; among them, 56 transcript levels were increased and another 53 were decreased in Ren(1a–9), Ren(2–9), or both overexpressing cell lines. As expected, there was a huge overlap of differentially expressed genes between Ren(1a–9) and Ren(2–9) cells. Slightly more than half of the identified genes were included in the AKT/mTOR signaling pathway, showing increased transcript levels in at least one renin-b-overexpressing cell line. The compact AKT/mTOR signaling pathway is illustrated as a schematic overview in Figure 6. Therein, proteins encoded by differentially expressed genes (*q* < 0.05, fold change ≥ |1.3|) in Ren(1a–9) (Figure 6A) and Ren(2–9) (Figure 6B) overexpressing cells are highlighted according to the transcriptome analyses data.

Genes with increased transcript levels in both renin-b-overexpressing cell lines were coding for insulin receptor (*Insr*), related RAS viral (*r-ras*) oncogene homolog (*Rras*), Son of sevenless homolog 1 (*Sos1*), SHC (Src homology 2 domain-containing)-transforming protein 1 (*Shc1*), v-akt murine thymoma viral oncogene homolog 1 (*Akt1*), glycogen synthase kinase 3 beta (*Gsk3b*), cyclin D1 (*Ccnd1*), cyclin-dependent kinase inhibitor 1A (*Cdkn1a*), MDM2 proto-oncogene, E3 ubiquitin protein ligase (*Mdm2*), mitogen-activated protein kinase 3 (*Mapk3*), regulatory associated protein of mTOR, complex 1 (*Rptor*), ras-related C3 botulinum toxin substrate 1 (*Rac1*), and unc-51-like autophagy-activating kinase 1 (*Ulk1*). Exclusively in Ren(1a–9)-overexpressing cells, we identified glycogen synthase kinase 3 alpha (*Gsk3a*), Bcl2-like 1 (*Bcl2l1*), and tuberous sclerosis 2 (*Tsc2*) to exhibit increased mRNA levels as compared with pIRES controls. However, in Ren(2–9)-overexpressing cells only, genes coding for insulin receptor substrate 1 (*Irs1*) and growth factor receptor-bound protein 2 (*Grb2*) were found with increased transcript levels (Figure 5 and Figure 6).

Furthermore, several genes were identified with decreased expression in both overexpressing groups. Among them were genes coding for B-cell CLL/lymphoma 2 (*Bcl2*), ribosomal protein S6 (*Rps6*), eukaryotic translation initiation factor 4 gamma 3 (*Eif4g3*), several eukaryotic translation initiation factor 3 subunits (*Eif3a*, *Eif3d*, *Eif3g*, *Eif3i*, *Eif3j*, and *Eif3m*), as well as numerous 40S ribosomal proteins (*Rps6*, *Rps8*, *Rps10*, *Rps18*, *Rps25*, and *Rps27l*). However, the eukaryotic translation initiation factor 4B-encoding gene *Eif4b* and the ribosomal protein subunits *Rps24* and *Rps26* were identified to show decreased mRNA levels in Ren(1a–9)-overexpressing cells only. At the same time, decreased transcript levels of DEP domain-containing mTOR-interacting protein (*Deptor*), mitogen-activated protein kinase-associated protein 1 (*Mapkap1*), eukaryotic translation-initiation factor 4E-binding protein 1 and 2 (*Eif4ebp1* and *Eif4ebp2*), eukaryotic translation-initiation factor 3 subunit C (*Eif3c*), and the ribosomal protein *Rps20* were detected exclusively in Ren(2–9)-overexpressing cells (Figure 5 and Figure 6).

Furthermore, a few differentially expressed genes were identified whose isoforms or subunits showed opposing directions of expression. These were genes encoding phosphoinositide-3-kinases (*Pik3c2b*, *Pik3r1,* and *Pik3r3*), protein phosphatases 2 (*Ppp2ca*, *Ppp2cb*, *Ppp2r1b*, *Ppp2r2a*, *Ppp2r2b*, *Ppp2r5a*, and *Ppp2r5b*), ras homolog family members A, C, G, and J (*Rhoa*, *Rhoc*, *Rhog*, and *Rhoj*), as well as protein kinases C, alpha, delta, eta, and iota (*Prkca*, *Prkcd*, *Prkch*, and *Prkci*) in at least one renin-b-overexpressing cell line (Figure 5 and Figure 6).

Since transcriptome data alone do not allow any conclusion about the activity of a system (increased transcript levels may stimulate the activity of a protein, but may also be the response to a decreased activity of this protein), we next asked whether AKT and mTOR were also altered with respect to abundance and activity, e.g., phosphorylation state on protein level.

### 3.4. Key Regulator Kinase AKT Is Activated by Renin-B

In both renin-b-overexpressing cell lines, the total protein level of AKT was unchanged compared to pIRES controls or slightly decreased (Figure 7A). However, residue serine 473 (Ser473) phosphorylated AKT (p-AKT), displaying the activated form of AKT, showed a significant increase in both renin-b-overexpressing cell lines (Figure 7B). Hence, the ratio of p-AKT to total AKT was increased in Ren(2–9) (*p* = 0.0042) and increased in trend in Ren(1a–9)-overexpressing cells (*p* = 0.0528) (Figure 7C).

Further analysis revealed that the total mTOR protein levels, the levels of phosphorylated mTOR at serine 2448 (p-mTOR), as well as the ratio of p-mTOR to total mTOR did not change significantly in renin-b-overexpressing cells as compared with pIRES controls (Figure 7D–F).

Overall, Western blot analyses demonstrated a significant increase in the phosphorylation of AKT (Ser473), providing independent support, in addition to the transcriptome data described above, that the AKT system is activated by renin-b overexpression (Figure 5 and Figure 6). Furthermore, the unaltered phosphorylation of mTOR (Ser2448) together with the increased *Rptor* and decreased *Deptor* transcript levels excludes suppression, but rather is pointing toward an activation or at least to a predisposition toward the activation of mTORC1.

### 3.5. Renin-B-Induced Warburg-like Effects Are Associated with Increased ROS Accumulation

Given the known role of the Warburg effect in regulating redox signaling [21], we next analyzed the mitochondrial and cytosolic accumulation of reactive oxygen species (ROS) (Figure 8). Using the MitoSOX Red mitochondrial superoxide indicator, the fluorescence intensity (FLI) of MitoSOX-positive cells was significantly increased in Ren(1a–9) and slightly, but not significantly, increased in Ren(2–9)-overexpressing cells, indicating an increased accumulation of mitochondrial superoxides, especially in Ren(1a–9) cells (Figure 8A). The FLI of another redox-sensitive fluorescence probe, dihydroethidium (DHE), which is oxidized by cytosolic and mitochondrial superoxides, as well as other ROS species, was unchanged in renin-b-overexpressing cell lines compared to pIRES controls (Figure. 8B). Concerning the dihydrorhodamine (DHR) labeling detecting cytosolic H_2_O_2_ and peroxynitrite anions, we observed a significant increase in DHR FLI only in Ren(1a–9)-overexpressing cells (Figure 8C).

The increased accumulation of ROS could be due to increased ROS production or decreased antioxidative mechanisms. Therefore, we re-analyzed our transcriptome data to identify differentially expressed genes coding for proteins that are involved in ROS generation or elimination. As illustrated in the heat map in Figure 8D, numerous transcripts of respiratory chain complexes I, II, and III coding for NADH dehydrogenase (ubiquinone), ubiquinol-cytochrome c reductase, and succinate dehydrogenase subunits were altered after the overexpression of renin-b in vitro. The vast majority of significantly altered genes involved in ROS production exhibited decreased transcript levels in both renin-b-overexpressing cell lines as compared with pIRES controls. These genes are coding for NADH dehydrogenase (ubiquinone) 1 alpha subcomplex 1 and 8 (*Ndufa1* and *Ndufa8*), NADH dehydrogenase (ubiquinone) complex I, assembly factor 5 (*Ndufaf5*), NADH dehydrogenase (ubiquinone) Fe-S protein 3 (*Ndufs3*), NADH dehydrogenase (ubiquinone) flavoprotein 3 and its pseudogene 1 (*Ndufv3 and Ndufv3-ps1*), ubiquinol-cytochrome c reductase core protein 1 (*Uqcrc1*), as well as succinate dehydrogenase complex, subunit D, integral membrane protein (*Sdhd*). However, decreased transcript levels of NADH dehydrogenase (ubiquinone) 1 alpha subcomplex 4 (*Ndufa4*) and ubiquinol-cytochrome c reductase hinge protein (*Uqcrh*) were exclusively detected in Ren(1a–9)-overexpressing cells, while in Ren(2–9)-overexpressing cells, only a reduced expression of genes coding for NADH dehydrogenase (ubiquinone) 1 alpha subcomplex 10-like1 (*Ndufa10l1*), NADH dehydrogenase (ubiquinone) 1 beta subcomplex 6 (*Ndufb6*), and NADH dehydrogenase (ubiquinone) complex I, assembly factor 4 (*Ndufaf4*) was observed.

Moreover, the gene expression of non-mitochondrial ROS generators, e.g., the transcript levels of cytosolic and microsomal xanthine dehydrogenase (*Xdh*), were increased in both renin-b-overexpressing cell lines, but increased gene expression of membrane-bound NADPH oxidase 1 (NOX1) occurred exclusively in Ren(2–9)-overexpressing cells. The transcript levels of NADPH oxidase members dual oxidase 1 and 2 (*Duox1* and *Duox2*) were decreased, while the mRNA levels of the endoplasmic reticulum oxidoreductase alpha and beta (*Ero1a* and *Ero1b*) were increased in both renin-b-overexpressing cell lines as compared with pIRES controls.

Cells are protected from ROS by multiple defense systems and antioxidants, such as superoxide dismutases (SOD), glutathione peroxidases (GPX), peroxiredoxins (PRDX), and thioredoxin reductases (TXNRD). Among these different ROS scavengers, we found increased transcript levels of *Sod3*, *Gpx4* (only in Ren(1a–9) cells), *Gpx8* (only in Ren(2–9) cells), *Prdx3*, *Prdx5*, *Txnrd1*, and *Txnrd3* (only in Ren(1a–9) cells), while the gene expression of glutathione reductase (*Gsr*) was reduced in Ren(2–9) cells.

Taken together, an increased accumulation of mitochondrial superoxides and cytosolic H_2_O_2_ in renin-b-overexpressing cells indicates that the cellular balance of ROS producers vs. ROS scavengers seems to be shifted toward an increased ROS production. However, the transcript levels of several genes encoding H_2_O_2_-degrading enzymes were found to be increased, and further analyses are needed to uncover the involvement of ROS production and scavengers.

## 4. Discussion

The discovery of a non-secretory cytosolic isoform of renin, termed renin-b, revealed several surprises. In contradiction to the classical secretory renin, which is known to exert harmful effects, renin-b was shown to be protective under starvation conditions. While the classical renin-angiotensin system increases ROS production, apoptosis, necrosis, inflammation, and fibrosis [22], our group has previously shown a reduced infarct size in isolated perfused hearts of transgenic rats overexpressing renin-b exposed to ischemia-reperfusion injury [4]. Furthermore, we observed increased resistance of primary cardiomyocytes from transgenic rats against glucose depletion-induced apoptosis and reduced necrosis and apoptosis in renin-b-overexpressing H9c2 cardiomyoblasts exposed to glucose and/or oxygen deprivation [5,6,13,14].

The mechanisms of renin-b action are unknown so far. Therefore, in this study, we aimed to define and characterize signaling pathways that unravel the renin-b effects placed between overexpression and its physiological function. Using transcriptome analysis, we were able to associate alterations in the transcript level with already known renin-b effects. From the multitude of affected genes, we focused on transcripts encoding kinases and substrates involved in the AKT/mTOR pathway, whose activation subsequently influences the transcript levels of glycolytic and TCA-cycle enzymes, as well as transporters associated with the Warburg effect. In addition, we considered transcripts coding for proteins associated with ROS generation or degradation. The obtained results of transcriptome analysis connected with Warburg-like effects are summarized in Figure 9.

Several studies have demonstrated that PI3K/AKT/mTOR signaling plays a central role in the initiation of cell death, maintenance of cell survival, cell cycle progression, proliferation, regulation of transcription, protein synthesis, and autophagy, as well as in influencing cell metabolism [23,24]. Focusing on the central role of AKT in this pathway, we observed increased expression of *Akt1* (Figure 5 and Figure 6) as well as increased phosphorylation of AKT1 at Ser473 in both renin-b-overexpressing cell lines (Figure 7B). This phosphorylation is induced by the kinase mammalian target of rapamycin complex 2 (mTORC2) [25]. Full AKT activation also requires the phosphorylation of AKT by 3-phosphoinositide-dependent protein kinase 1 (PDPK1). PDPK1, together with mTORC2, is involved in the growth factor/PI3K signaling. Thus, these phosphorylation events enhance AKT kinase activity, allowing AKT to phosphorylate several substrates and facilitating the phosphorylation of tuberous sclerosis 2 (TSC2), for example, which in turn leads to the activation of mTORC1. Lastly, this cascade leads to a series of different effects culminating in the induction of protein synthesis [26], the phosphorylation of murine double minute 2 (MDM2), leading to p53 destabilization and thus preventing cell death [27], the activation of the transcription factor NF-κB resulting in the induction of prosurvival genes [28], the phosphorylation of glycogen synthase kinase 3 (GSK3) involved in metabolic processes and the regulation of the cell cycle [29], and the phosphorylation of the AKT substrate of 160 kDa (AS160/TBC1D4), facilitating GLUT4 translocation and glucose uptake [30].

Here, we demonstrate that the expression of the corresponding genes coding for TSC2, MDM2, GSK3, and GLUT4 was upregulated at least in one renin-b-overexpressing cell line (Figure 5). Furthermore, the transcript levels of *Rptor* encoding a factor necessary for mTORC1 activity and part of the mTORC1 complex were increased in both renin-b-overexpressing cell lines, while the transcript level of *Deptor* encoding an inhibitory factor of mTORC1 and mTORC2 was decreased in Ren(2–9)-overexpressing cells only (Figure 5). Surprisingly, in both renin-b-overexpressing cell lines, transcripts coding for components of the 40S ribosome subunit-mRNA complex (RPS6, ElF3, and the 40S ribosome subunits itself) as part of the mTORC1 downstream responses showed decreased levels (Figure 5), indicating reduced mRNA translation, protein synthesis, and cell proliferation. Therefore, the stimulation of AKT/mTOR signaling must be taken as a response counteracting this downregulation. Taken together, our data support the hypothesis of the activation of the PI3K/AKT/mTOR pathway by renin-b on the level of increased gene expression, as well as AKT activation (Figure 5, Figure 6, and Figure 7). Although the primary event remains unknown, the activation of this pathway might contribute to the metabolic alterations induced by renin-b and primes cells for better survival under starvation conditions.

Despite increased gene expression and activation of AKT1, previous studies from our group detected an increased basal apoptosis rate in both renin-b-overexpressing H9c2 cell lines [6,13], indicating a potential disadvantage of the chronic stimulation of AKT1. Indeed, the chronic activation of AKT1 in the heart results in larger infarct areas and poor recovery in mice exposed to ischemia-reperfusion injury [31]. On the other hand, Kunuthur et al. [32] showed that AKT1 is a mediator of ischemic preconditioning associated with cardiac protection. Thus, mice deficient in AKT1 were unable to inactivate GSK3B and activate the ERK1/2 pathways that are necessary for protection against ischemia-reperfusion injury. However, the pro-apoptotic effects of renin-b observed in cell culture appear to be without significance in vivo, since transgenic rats overexpressing renin-b do not develop any pathological phenotype [33]. On the other hand, the protective effects of renin-b under starvation conditions observed in cell culture are well reflected by the reduced infarct size in isolated perfused hearts of renin-b-transgenic rats [4]. Because we detected increased expression of *Akt1*, *Gsk3b,* and *Mapk3* (ERK1) (Figure 5), we speculate that the previously observed protection by renin-b against oxygen and glucose deprivation, as well as ischemia-reperfusion injury, may be mediated in this manner. Additionally, another factor that may be involved is insulin. Insulin, when given prior to ischemia or at reperfusion, can protect the heart from ischemia-reperfusion injury, as evidenced by the reduced infarct size [34,35]. Thus, the increased mRNA levels of insulin receptor (*Insr*) (Figure 5) observed in both renin-b-overexpressing H9c2 cells have the potential to initiate the activation of the PI3K/AKT/mTOR signaling cascade, resulting in AKT-mediated cardioprotective effects during ischemia-related conditions.

In addition, AKT2 plays a key role in PI3K signaling by controlling glucose metabolism, including glycogenolysis, gluconeogenesis, and glucose receptor translocation as part of the signal transduction downstream of the insulin receptor [36]. Among others, AKT2 regulates glucose uptake by mediating the activation of insulin receptor substrate (Irs) and Rho family small GTPase Ras-related C3 botulinum toxin substrate 1 (RAC1), leading to the translocation of glucose transporters GLUT1 and GLUT4, encoded by *Slc2a1* and *Slc2a4*, to the cell surface and subsequently to increased glucose uptake [37,38]. Interpreting our transcriptome data, we hypothesize that increased *Insr* expression and signaling are associated with the PI3K-dependent activation of AKT2 and RAC1, leading to both increased gene expression and translocation of GLUT4, and finally to the detected increased glucose uptake (Figure 4). Interestingly, the (pro)renin receptor (ATP6AP2), which is known to bind (pro)renin, activates RAC1 as well. Thus, the (pro)renin receptor binds partitioning defective 3 homolog protein (PARD3), thereby activating atypical protein kinase C (aPKC) and, subsequently, RAC1 [39]. Taken together, the interaction of renin-b with ATP6AP2 could be responsible for increased glucose uptake. In support, our transcriptome data show increased expression of *Pard3* (data not shown) and *Rac1* (Figure 5) in renin-b-overexpressing cell lines.

While the *Slc2a4*-encoded GLUT4 is an insulin-dependent transporter, the *Slc2a1*-encoded GLUT1 uniporter is responsible for basal insulin-independent glucose uptake [40,41]. Cardiac-specific overexpression of GLUT1 increases glucose uptake, glycolysis, and the accumulation of glycogen storage in the postnatal heart. This enhanced glucose metabolism then promotes neonatal heart regeneration and inhibits fibrosis upon cryoinjury [42], indicating that cardioprotective effects could also be mediated by increased GLUT1 gene expression. Transcription factors SP1 and SP3 regulate *Slc2a1* expression in the heart. While SP1 stimulates the transcription of *Slc2a1* by binding the promoter region, SP3 acts as a repressor [43]. Because the expression of *Sp3* is reduced in both renin-b-overexpressing cells (data not shown), the SP1 effect could dominate and, therefore, may lead to the detected increased *Slc2a1* expression (Figure 4).

An increased uptake of glucose further requires an enhanced flux of glucose through the glycolysis pathway and, therefore, the activation of different glycolytic enzymes, such as hexokinase 2 (HK2) [44], phosphofructokinase 1 (PFK1) [45], or lactate dehydrogenase A (LDHA) [46], which can be mediated by the kinases AKT and mTOR. Although we did not analyze the phosphorylation state of glycolytic enzymes, we found increased transcript levels of a series of genes coding for glycolytic enzymes (*Hk1*, *Pfkm*, *Pfkfb3*, *Pgk1*, and *Pgam2*), as illustrated in the glycolysis heat map (Figure 2), indicating an enhanced flux of glucose through the glycolysis pathway in renin-b-overexpressing cells. In tumors, but also in proliferating cells, the enhanced flux through the glycolysis pathway directs pyruvate to lactate catalyzed by LDHA. This results in an increased output of lactate via monocarboxylate transporter 4 (MCT4 encoded by *Slc16a3*), followed by the extracellular accumulation of lactate. Indeed, we found an increased extracellular lactate content and a partly reduced glucose/lactate ratio, indicating a forced conversion of glucose to lactate in renin-b-overexpressing cells (Figure 4). Although the transcript levels of genes coding for LDHA and MCT1-4 were unaltered, we interpreted the increased glucose uptake and lactate efflux together with the increased expression of genes coding for glycolytic enzymes as a possible metabolic switch to enhanced aerobic glycolysis, known as the Warburg effect.

MicroRNAs (miRNAs) are a family of functional RNAs involved in the post-transcriptional regulation of gene expression. Several miRNAs contribute to regulating the Warburg effect. Warburg effect-relevant targets of these miRNAs are especially mRNAs coding for glucose transporters, glycolytic enzymes (HK, PFK, PKM2, LDHA), PI3K and PDPK1, AKT, mTORC1, p53, and hypoxia-inducible factor 1 (HIF1) [47,48]. Focusing on certain Warburg effect-relevant miRNAs, we found a decreased level of miR-1291 in Ren(1a–9) cells. Because miR-1291 is known to regulate the expression of *Slc2a1* [49], decreased miR-1291 in renin-b-overexpressing cells may contribute to the increased expression of *Slc2a1* coding for GLUT1. Additionally, miR-221 and miR-21 function as downstream activators of AKT through the suppression of the phosphatase and tensin homolog (PTEN) that inhibits AKT [50,51]. Because the levels of miR-221 and miR-21 were increased in renin-b-overexpressing cells, they may have contributed to the increased transcript levels of *Akt1*. Another markedly altered miRNA was miR-let-7f-1, showing increased expression in renin-b-overexpressing cells (Figure 3). By inhibiting mTORC1 signaling, miR-let-7f-1 increases autophagy and, therefore, cellular survival during starvation conditions [48]. Indeed, in renin-b-overexpressing cells, the transcript levels of *Ulk1* coding for an autophagy-related protein that is located downstream of mTORC1 were increased, indicating miR-let-7f-1-induced inhibition of mTORC1. Therefore, we conclude that the AKT-mediated activation of mTORC1 could be counteracted via miR-let-7f-1.

Another main adaptive metabolic response mediating the Warburg effect is the disengagement of glycolysis from the mitochondrial TCA cycle. Here, we hypothesize that the increased expression of *Pdk2* coding for mitochondrial pyruvate dehydrogenase kinase 2 by inhibiting the activity of the pyruvate dehydrogenase (PDH) complex contributes to the decreased entry of pyruvate into the TCA cycle. Indeed, the *Pdk2* transcript levels were markedly increased in both H9c2 cell lines overexpressing renin-b (Figure 2). This suggests that, in renin-b overexpressing cells, increased *Pdk2* expression could represent the event initializing the Warburg effect and finally lead to the observed extracellular accumulation of lactate, as well as increased glucose uptake. Considering the transcript levels of TCA-cycle enzymes in this scenario, we found decreased levels of transcripts coding for isocitrate dehydrogenases 3 alpha (*Idh3a*) and beta (*Idh3b*), succinate-CoA ligase ADP-forming subunit beta (*Sucla2*), and GDP-forming subunit beta (*Suclg2*), as well as succinate dehydrogenase complex subunit D (*Sdhd*) (Figure 2). The reduced expression would result in increased accumulation of TCA cycle metabolites, such as *α*-ketoglutarate, succinate, or fumarate, within the mitochondrial matrix, finally resulting in their increased release into the cytosol. Indeed, the transcript levels of different soluble carrier family 25 members (*Slc25a1*, *Slc25a10*, and *Slc2511*) coding for different mitochondrial carriers transporting citrate, dicarboxylate, and oxoglutarate were increased in renin-b-overexpressing cells (Figure 2). It is known that metabolites, such as fumarate, succinate, and lactate, can inhibit prolyl hydroxylases that are involved in the degradation of HIF1A [52,53,54]. By inhibiting these prolyl hydroxylases, HIF1A becomes stabilized, thus activating the HIF pathway, even under aerobic conditions. The HIF pathway, as being involved in the Warburg effect, redirects energy production from mitochondria toward glycolysis [52,55]. Furthermore, HIF1A is stabilized by enhanced PI3K signaling [56,57], as well as by increased levels of ROS and nitric oxide (NO) [58], which were indeed detected in our renin-b-overexpressing cells (Figure 8). However, the transcript levels of *Hif1a* itself were unchanged in our study and we did not analyze the HIF1A protein abundance.

Highlighting the roles of oxoglutarate dehydrogenase (lipoamide) (OGDH) (Figure 2) and succinate dehydrogenase complex D (SDHD) (Figure 8D), whose transcript levels showed significant alterations (increased and decreased, respectively) in renin-b-overexpressing cells, we suggest that these dehydrogenases could be involved in the detected increase in mitochondrial superoxide production. Recent studies have shown that, apart from complex I and III of the respiratory chain, oxoglutarate dehydrogenase, pyruvate dehydrogenase, and succinate dehydrogenase can serve as sources of mitochondrial superoxides and H_2_O_2_ [59,60,61]. Xiao et al. [62] demonstrated that a decrease in the transcript levels of *Sdhc* and *Sdhd* coding for the components of complex II of the respiratory chain is associated with a decrease in complex II activity and an increase in the mitochondrial ROS levels. Therefore, the renin-b-induced increase in the mitochondrial superoxide levels could be due to the reduced expression of *Sdhd,* indicating enhanced levels of oxidative stress, as well as confirming the potential stabilizing effect on HIF. In addition, the PI3K/AKT signaling pathway itself is involved in NADPH oxidase (NOX)-derived cytosolic ROS production [63]. Upon activation, the cytosolic subunits of NOX interact with integral membrane subunits, forming functional NOX enzymes that, in turn, generate ROS [64]. PI3K/AKT inhibitors can reduce this NOX-dependent ROS generation through the inhibition of NOX subunit translocation into the membrane. Moreover, oxidative stress inhibits PTEN-induced PI3K/AKT signaling, promoting both the expression of cell-survival genes, as well as further ROS production [65]. By increasing the flux of glucose into the pentose phosphate pathway, cells can counteract increased ROS production by the enhanced generation of reducing equivalents (NADPH and glutathione). Stimulated glucose metabolism in response to oxidative stress is indeed assumed to increase the generation of reducing equivalents to detoxify H_2_O_2_ generated from superoxides by superoxide dismutase. Accordingly, we detected increased transcript levels of superoxide dismutase 3 (*Sod3*), glutathione peroxidases 4 (*Gpx4*) and 8 (*Gpx8*), as well as peroxiredoxins 3 (*Prdx3*) and 5 (*Prdx5*) (Figure 8D).

Taken together, our study provides evidence for a complex renin-b-induced mechanism, including the activation of the PI3K/AKT/mTOR signaling pathway and enhanced ROS accumulation, as shown in Figure 9. This may lead to the manifestation of a Warburg-like phenotype mediating the reduced responsiveness to harmful effects during stress conditions. Currently, we do not know whether the slightly different profiles of Ren(1a–9) and Ren(2–9)-overexpressing cells can be attributed to a role of the 5′UTR, to different levels of renin-b overexpression, or to other factors. We are aware of the fact that our study was performed with a cell line only, which does not represent the in vivo situation of adult cardiomyocytes in terms of proliferation, metabolism, and signaling. Therefore, the data obtained should be used with caution.

What would be the implications of our findings? Renin-b overexpression induces a variety of metabolic alterations, such as increased expression of genes involved in glucose transport, glycolysis, mitochondrial transport, and ROS generation, together with decreased expression of genes coding for TCA-cycle components and genes encoding proteins involved in translation. Therefore, the expression of renin-b should be rather disadvantageous under healthy conditions. However, this is in agreement with the fact that renin-b expression is extremely low under healthy conditions, and is only induced under starvation or otherwise challenging conditions. Moreover, transgenic rats overexpressing renin-b do not exhibit any pathological phenotype [33]. Our studies indicate that it is advantageous to induce mild or moderate overexpression of renin-b prior to starvation to be able to handle this challenge more rapidly and efficiently.

To confirm our data and attribute them indeed to the overexpression of renin-b, it would be desirable to perform additional experiments showing that the downregulation of renin-b is able to reverse the effects of renin-b overexpression. Furthermore, the role of the downregulation of renin-b in otherwise untreated cells would be of interest.

## 5. Summary and Conclusions

The present study indicates, with respect to metabolic changes, that renin-b overexpression induces an upregulation of aerobic glycolysis known as the Warburg effect, a response similar to metabolic remodeling under starvation conditions. This conclusion is derived from transcriptome profiling and functional data comprising (I) increased expression of genes encoding glucose transporters combined with increased glucose consumption; (II) increased transcript levels of numerous genes coding for enzymes involved in glycolysis; (III) increased pyruvate dehydrogenase kinase 2 (*Pdk2*) transcript level, whose encoded protein suppresses pyruvate entry into the TCA cycle; (IV) reduced transcript levels of several genes coding for TCA cycle enzymes; (V) increased accumulation of reactive oxygen species induced by respiratory chain complexes and NADPH oxidase; and (VI) induction of PI3K/AKT/mTOR signaling sustaining enhanced glycolysis.

We suggest that renin-b plays an essential physiological role under starvation conditions, where the PI3K/AKT/mTOR signaling cascade is known to be activated and renin-b gene expression is increased as observed after cardiac infarction in vivo. This study also provides an extensive set of data to generate hypotheses with respect to the importance of renin-b, the Warburg effect, and cell survival.

## Figures and Tables

**Figure 1 cells-11-01459-f001:**
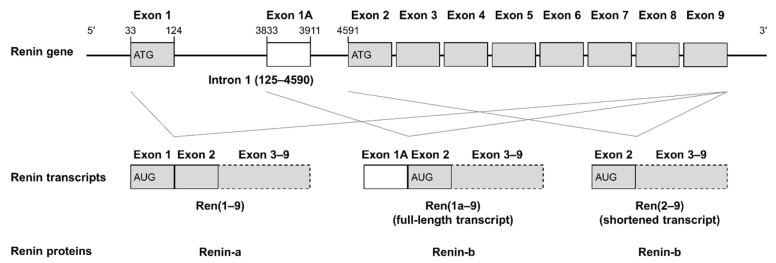
Structure of the rat renin gene and transcripts encoding two different renin proteins. The renin gene comprises 9 exons (gray boxes), where an alternative exon 1 named exon 1A (white box) is located within intron 1. Transcription leads to three transcript variants comprising *Ren(1–9),* including the classical exon 1 as well as the alternative transcripts *Ren(1a–9)* and *Ren(2–9)*, where *Ren(1a–9)* comprises the 5′ UTR located in exon 1a and *Ren(2–9)* is missing it. The transcription of *Ren(1–9)* starts at AUG in exon 1, while for *Ren(1a–9)* and *Ren(2–9),* the start codon is located in exon 2, resulting in either the classical secretory renin-a or the alternative non-secretory renin-b.

**Figure 2 cells-11-01459-f002:**
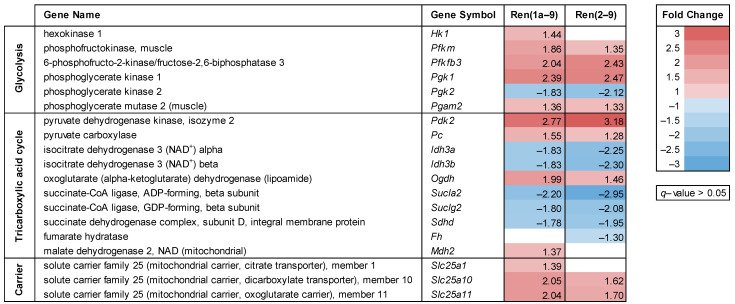
Alterations in the expression of genes involved in glycolysis, the tricarboxylic acid (TCA) cycle, and mitochondrial transport. The transcript levels were identified by microarray analyses as a result of either Ren(1a–9) or Ren(2–9) overexpression as compared with pIRES H9c2 control cells. The red color indicates genes with increased and the blue color indicates those with decreased mRNA levels as compared with pIRES controls, while the white color indicates genes that were not significantly altered with *q* > 0.05. Shown are only significantly altered genes (*q* < 0.05) that exhibit a fold change ≥ |1.3| in at least one group comparison.

**Figure 3 cells-11-01459-f003:**
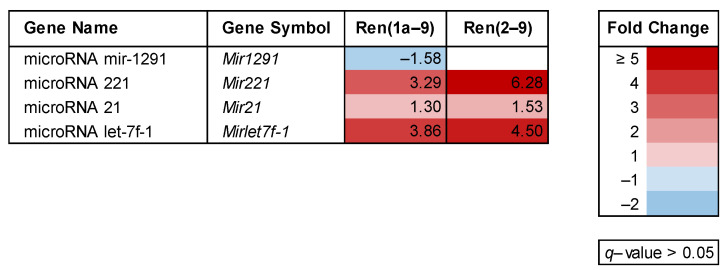
Alterations in the expression of genes coding for microRNAs. The transcript levels were identified by microarray analyses as a result of either Ren(1a–9) or Ren(2–9) overexpression as compared with pIRES H9C2 control cells. The red color indicates genes with increased mRNA levels and the blue color indicates those with decreased mRNA levels as compared with pIRES controls, while the white color indicates not significantly altered genes with *q* > 0.05. Shown are only significantly altered genes (*q* < 0.05) of selected miRNAs involved in glucose uptake or AKT/mTOR signaling.

**Figure 4 cells-11-01459-f004:**
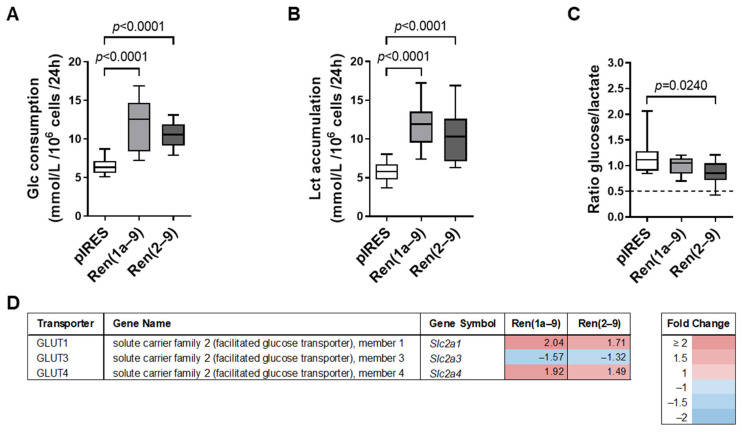
Metabolic alterations induced by renin-b overexpression. (**A**) Increased glucose (Glc) consumption, (**B**) increased extracellular lactate (Lct) accumulation, and (**C**) decreased ratio of glucose consumption to lactate accumulation as a result of either Ren(1a–9) or Ren(2–9) overexpression as compared with pIRES H9c2 control cells. The dashed line indicates the Glc/Lct ratio at anaerobic glycolysis when glucose is completely converted to lactate. (**D**) Expression of significantly altered class I glucose transporter-coding genes as identified by transcriptome analyses (*q* < 0.05). The red color indicates genes with increased mRNA levels and the blue color indicates those with decreased mRNA levels as compared with pIRES controls. (**A**–**C**) Data of 9–18 independent experiments per group are shown as box plots, where the top of the box represents the 75th percentile, the bottom of the box represents the 25th percentile, and the line in the middle represents the median. The whiskers represent the highest and lowest values in the dataset. Data were analyzed by one-way ANOVA followed by Tukey’s test. Single *p*–values < 0.05 vs. pIRES controls are given in the figure.

**Figure 5 cells-11-01459-f005:**
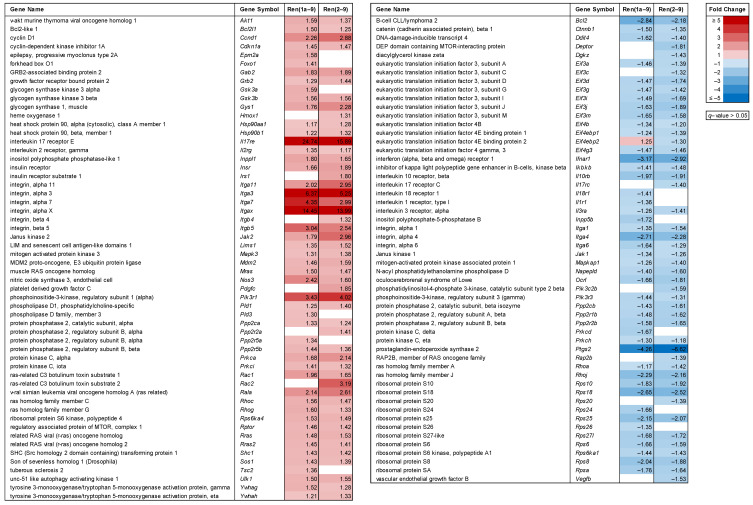
Alterations in the expression of genes involved in AKT/mTOR signaling. The transcript levels were identified by microarray analyses as a result of either Ren(1a–9) or Ren(2–9) overexpression as compared with pIRES H9C2 control cells. The red color indicates genes with increased mRNA levels and the blue color indicates those with decreased mRNA levels as compared with pIRES controls, while the white color indicates not significantly altered genes with *q* > 0.05. Shown are only significantly altered genes (*q* < 0.05) that exhibit a fold change of ≥ |1.3| in at least one group comparison.

**Figure 6 cells-11-01459-f006:**
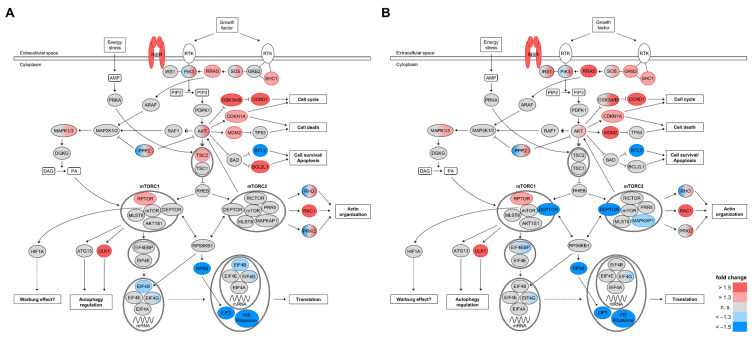
Schematic overview of the AKT/mTOR signaling pathway. Protein encoding genes identified with differential expression (*q* < 0.05, fold change ≥ |1.3|) in either (**A**) Ren(1a–9) or (**B**) Ren(2–9) overexpressing H9c2 cells as compared with pIRES controls are highlighted. The red color indicates genes with increased mRNA levels and the blue color indicates those with decreased mRNA levels as compared with pIRES controls, while the gray color indicates not significantly altered genes with *q* > 0.05 or fold change < |1.3|. Gene expression data and the schematic pathway were based on microarray analyses and Ingenuity pathway analysis (IPA) software, respectively. AKT: v-akt murine thymoma viral oncogene homolog/protein kinase B; AKT1S1: AKT1 substrate 1 (proline-rich); ARAF: A-Raf proto-oncogene, serine/threonine kinase; ATG13: autophagy-related 13; BAD: BCL2-associated agonist of cell death; BCL2: B-cell CLL/lymphoma 2; BCL2L1: Bcl2-like 1; CCND1: cyclin D1; CDKN1A: cyclin-dependent kinase inhibitor 1A; DAG: diacylglycerol; DEPTOR: DEP domain containing mTOR-interacting protein; DGKG: diacylglycerol kinase, gamma; EIF3: eukaryotic translation initiation factor 3; EIF4A: eukaryotic translation initiation factor 4A; EIF4B: eukaryotic translation initiation factor 4B; EIF4E: eukaryotic translation initiation factor 4E; EIF4EBP: eukaryotic translation initiation factor 4E binding protein; EIF4G: eukaryotic translation initiation factor 4, gamma; GRB2: growth factor receptor-bound protein 2; GSK3: glycogen synthase kinase 3; HIF1A: hypoxia-inducible factor 1, alpha subunit; INSR: insulin receptor; IRS1: insulin receptor substrate 1; MAP2K1/2: mitogen-activated protein kinase kinase 1/2; MAPK1/3: mitogen-activated protein kinase 1/3; MAPKAP1: mitogen-activated protein kinase associated protein 1; MDM2: MDM2 proto-oncogene, E3 ubiquitin protein ligase; MLST8: mTOR-associated protein, LST8 homolog; mTOR: mechanistic target of rapamycin (serine/threonine kinase); PA: phosphatidic acid; PDPK1: 3-phosphoinositide-dependent protein kinase-1; PIK3: phosphoinositide-3-kinase; PIP2: phosphatidylinositol-4, 5-bisphosphate; PIP3: phosphatidylinositol-3, 4, 5-triphosphate; PPP2: protein phosphatase 2; PRKA: protein kinase, AMP-activated; PRKC: protein kinase C; PRR5: proline rich 5 (renal); RAC1: ras-related C3 botulinum toxin substrate 1; RAF1: v-raf-leukemia viral oncogene 1; RHEB: ras homolog enriched in brain; RHO: ras homolog family member; RICTOR: RPTOR-independent companion of mTOR, complex 2; RPS6: ribosomal protein S6; RPS6KB1: ribosomal protein S6 kinase; RPTOR: regulatory associated protein of mTOR, complex 1; RRAS: related RAS viral (r-ras) oncogene homolog; RTK: Receptor tyrosine kinase; SHC1: SHC (Src homology 2 domain-containing) transforming protein; SOS: Son of sevenless homolog; TP53: tumor protein p53; TSC1: tuberous sclerosis 1; TSC2: tuberous sclerosis 2; ULK1: unc-51-like autophagy-activating kinase 1.

**Figure 7 cells-11-01459-f007:**
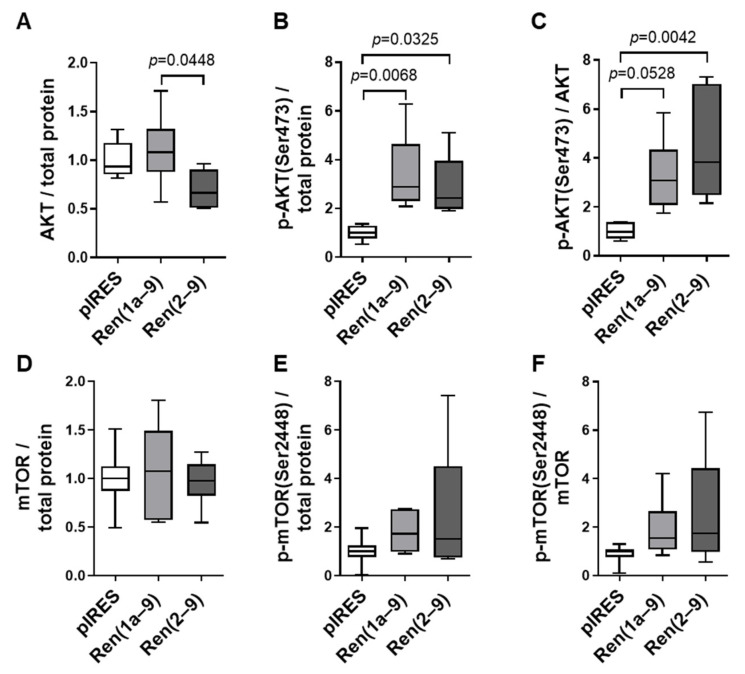
Protein abundances of the kinases AKT and mTOR. The protein levels of total and phosphorylated (**A**–**C**) AKT and (**D**–**F**) mTOR were detected by Western blot analysis of a pIRES control as well as Ren(1a–9) and Ren(2–9)-overexpressing H9c2 cells and normalized to total protein content. Representative western blot images of the protein abundances of the kinases AKT and mTOR are provided in Appendix A. Data of 6 independent experiments per group are shown as box plots, where the top of the box represents the 75th percentile, the bottom of the box represents the 25th percentile, and the line in the middle represents the median. The whiskers represent the highest and lowest values in the dataset. The data were analyzed by one-way ANOVA followed by Tukey’s test. Single *p*–values < 0.1 vs. pIRES controls or as indicated are given in the figure.

**Figure 8 cells-11-01459-f008:**
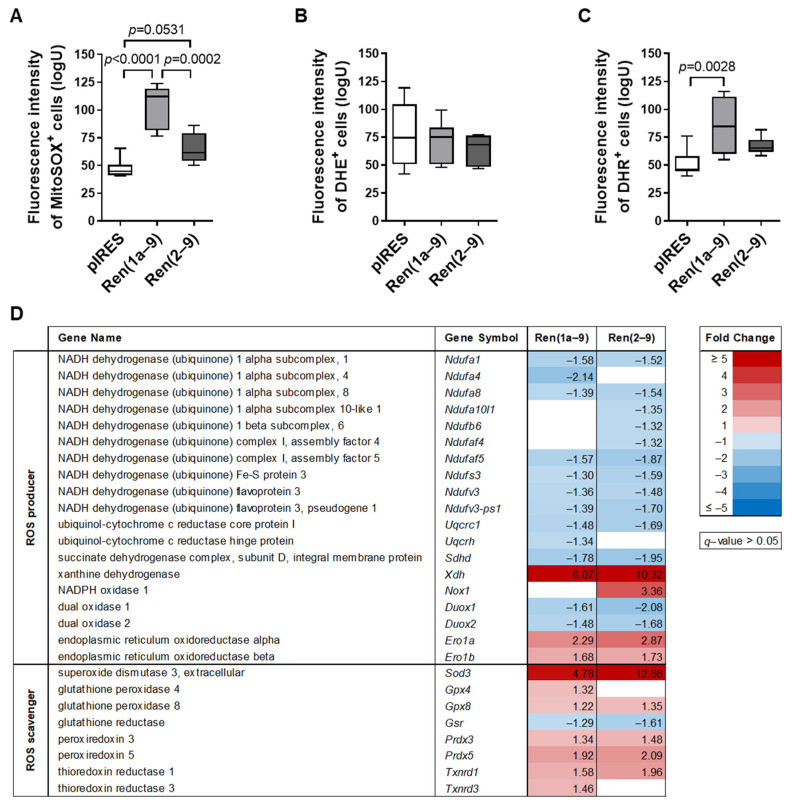
Renin-b-induced alterations in mitochondrial and cytosolic ROS accumulation. (**A**) Increased fluorescence intensity (FLI) of MitoSOX-positive cells, (**B**) unchanged FLI of dihydroethidium (DHE)-positive cells, and (**C**) increased FLI of dihydrorhodamine (DHR)-positive cells as a result of either Ren(1a–9) or Ren(2–9) overexpression as compared with pIRES H9c2 control cells. (**D**) Alterations in the expression of genes coding for reactive oxygen species producers or scavengers as identified by transcriptome analyses. Shown are only significantly altered genes (*q* < 0.05) that exhibited a fold change of ≥ |1.3| in at least one group comparison. (**A**–**C**) Data of 7–9 independent experiments per group are shown as box plots, where the top of the box represents the 75th percentile, the bottom of the box represents the 25th percentile, and the line in the middle represents the median. The whiskers represent the highest and lowest values in the dataset. The data were analyzed by one-way ANOVA followed by Tukey’s test. Single *p*–values < 0.1 vs. pIRES controls or as indicated are given in the figure.

**Figure 9 cells-11-01459-f009:**
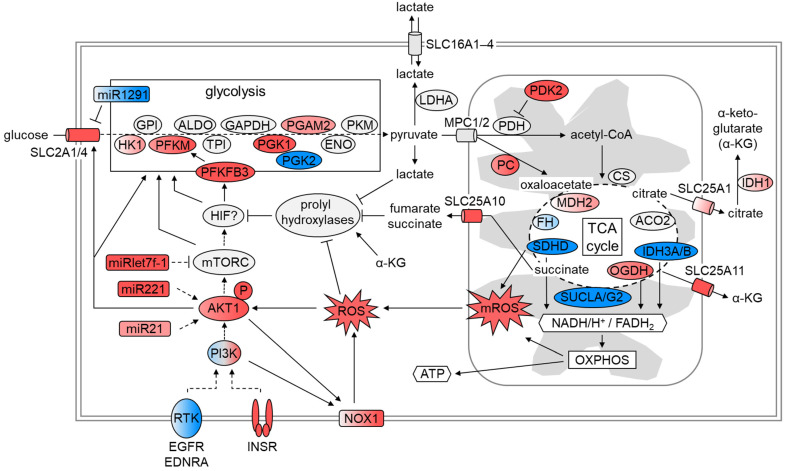
Molecular changes driving the Warburg effect. Downstream from the insulin receptor (INSR), phosphatidylinositol 3-kinase (PI3K) activates protein kinase B (AKT) stimulating the mechanistic target of rapamycin complex (mTORC), a protein kinase complex altering metabolism directly or via hypoxia-inducible factor 1 (HIF-1). mTORC and HIF-1 stimulate the expression of glucose transporters (GLUT), glycolytic enzymes, and pyruvate dehydrogenase kinase (Pdk2) that blocks the pyruvate dehydrogenase complex (Pdh). By blocking Pdh, the entry of pyruvate into the tricarboxylic acid (TCA) cycle is reduced. Alternatively, pyruvate is catalyzed by lactate dehydrogenase A (Ldha) to lactate. Lactate stabilizes HIF-1, whose accumulation leads to the increased gene expression of glycolytic enzymes and glucose transporters. Lactate after efflux via monocarboxylate transporter 4 (MCT4) accumulates in the cell supernatant. Additionally, the expression of several genes encoding TCA enzymes is reduced, leading to the accumulation of several TCA substrates in the cytosol that, again, can influence the stability of HIF-1. The red color indicates genes with increased mRNA levels and the blue color indicates those with decreased mRNA levels as compared with pIRES controls (*q* < 0.05, fold change ≥ |1.3|), while the gray color indicates not significantly altered genes with *q* > 0.05 or fold change < |1.3|.

**Table 1 cells-11-01459-t001:** Primer sequences for the detection of transcript abundances.

Transcript	Forward Primer (5′ to 3′)	Reverse Primer (5′ to 3′)
*exon(1a–9)*	exon 1a: TGAATTTCCCCAGTCAGTGAT	exon 2: GAATTCACCCCATTCAGCAC
*exon(2–9)*	exon 6: GCTCCTGGCAGATCACCAT	exon 8: CCTGGCTACAGTTCACAACGTA
*Ywhaz*	exon 2–3: CATCTGCAACGACGTACTGTCTCT	exon 3–4: GACTGGTCCACAATTCCTTTCTTG

## Data Availability

Not applicable.

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
