# Peer review of "Overexpression of Renin-B Induces Warburg-like Effects That Are Associated with Increased AKT/mTOR Signaling"

_cells, 2022, doi:10.3390/cells11091459_

Round 1

Reviewer 1 Report

The manuscript 'Overexpression of renin-b induces Warburg-like effects that are associated with increased AKT/mTOR signaling' was an interesting paper looking at the pathways altered by overexpression of renin-b isoform, primarily using microarray analysis with some protein and functional data to support the gene analysis.  The data is clearly presented with the use of pathway diagrams to illustrate to the reader the changes detected. 

Comments to be addressed:

  1. The paper uses over expression of two plasmids encoding renin-b, one with the 5'UTR and one without. It would be useful to include a diagram illustrating the difference between the two constructs showing the same protein is encoded by both. 
  2. There is a difference in the gene expression of these two constructs by real-time PCR, is this significant? and does this translate to differences in protein expression?

Author Response

Response to reviewer 1

Comments to be addressed:

  1. The paper uses over expression of two plasmids encoding renin-b, one with the 5'UTR and one without. It would be useful to include a diagram illustrating the difference between the two constructs showing the same protein is encoded by both. 

Thank you for this suggestion. We have now included a figure illustrating the structure of the two different renin transcript variants in the methods part (please see figure 1 on page 3).

  1. There is a difference in the gene expression of these two constructs by real-time PCR, is this significant? and does this translate to differences in protein expression

This is a good question. However, our data do not indicate such differences. There is a 2.06- and 1.74-fold overexpression of Ren(1a-9) and Ren(2-9), respectively. Moreover, expression of Ren(1a-9) cannot be compared with expression of Ren(2-9) since different primer pairs were used that may have different efficiencies. With respect to renin protein levels in tissues other than the kidney, many investigators have failed due to unreliable antibodies. Therefore, we cannot provide renin protein data.

Reviewer 2 Report

In the manuscript entitled “Overexpression of renin-b induces Warburg-like effects that are associated with increased AKT/mTOR signaling” authors, Golchert et al, overexpressed renin b in H9c2 cells and used microarray-based transcriptome analyses to delineate signaling pathways involved in renin-b mediated cardio protective effects. By transcriptome profiling they observed an increase in glycolytic enzymes and glucose transporters genes while, TCA cycle related genes were suppressed.  Additionally, in renin-b overexpressed cells they observed an increase in glucose consumption and lactate accumulation which was accompanied by upregulation in AKT/mTOR signaling and ROS production. Based on these findings the authors claim that overexpression of renin-b induces Warburg effects in H9c2 cells. Overall, the results support the conclusions, however, there are some major and minor concerns.

Major weakness

  1. This study was performed in a cell-line, which is far away from the primary cardiomyocytes and in vivo whole body.
  2. The experiments are performed on an immortalized undifferentiated rat myofibroblasts, H9C2, and therefore the rate of proliferation, metabolism, and signaling of these cells may differ from primary cells.
  3. Explanations in material and methods are too short, detailed methods for overexpression of renin transcripts and transcriptome analysis should be included
  4. What is the major source for ROS accumulation? The authors may present data on NADPH oxidase activity and mitochondrial ROS generation in renin-b overexpressed H9c2 cells

Minor weakness

  1. In supplementary file the names of the samples need to corrected.
  2. Did the authors look for the effects of renin-b expression on gene involved in mitochondrial dysfunction?
  3. The discussion is too, it should rather focus on main findings in the study.
  4. The conclusion section of the manuscript should be revised and avoid presenting a reference in conclusion.

Author Response

Response to reviewer 2:

Major weakness

  1. This study was performed in a cell-line, which is far away from the primary cardiomyocytes and in vivo whole body.
  2. The experiments are performed on an immortalized undifferentiated rat myofibroblasts, H9C2, and therefore the rate of proliferation, metabolism, and signaling of these cells may differ from primary cells.

We agree that our cell model has disadvantages and thank the reviewer for this hint.

We have now discussed this matter in the discussion section on page 21 in lines 715-718.

  1. Explanations “in material and methods are too short, detailed methods for overexpression of renin transcripts and transcriptome analysis should be included

We have now included a figure showing detailed structure of the two Ren(1a-9) and Ren(2-9) transcripts. Furthermore, we described the method of overexpression in more detail on page 3 in lines 99-108. The transcriptome analysis has already been described in detail in chapter 2.3 and in our opinion does not require any further details.

  1. What is the major source for ROS accumulation? The authors may present data on NADPH oxidase activity and mitochondrial ROS generation in renin-b overexpressed H9c2 cells

Presently, we cannot define the compartments and enzymes responsible for ROS production. However, it is well accepted that mitochondria are the main producers of superoxides and hydrogen peroxides depending on substrate catabolism and electron chain activity. In this context, mitochondrial matrix localized hydrogen peroxide can readily escape to the cytosol leading to the increased fluorescence intensity of DHR-positive cells obtained from renin-b overexpressing cell lines. However, we also found changed NADPH expression levels that could contribute to the increased fluorescence intensity of DHR-positive cells. We thank the reviewer for this interesting question and will investigate these aspects in future experiments.

Minor weakness

In supplementary file the names of the samples need to corrected.

We have carefully checked the supplementary table S1 again and have not found any mistakes.

  1. Did the authors look for the effects of renin-b expression on gene involved in mitochondrial dysfunction?

Mitochondrial dysfunction can result from an inadequate number of mitochondria, an inability to transport necessary metabolites into mitochondria, or a dysfunction in cellular respiration, particularly in the electron transport chain. Indeed, in renin-b overexpressing cells, we found altered expression of several genes coding for electron transport chain members. However, former studies have shown, that both the basal oxygen consumption rate as well as the ATP content of renin-b overexpressing cells were unchanged compared to controls.

  1. The discussion is too, it should rather focus on main findings in the study.

In the discussion, we have already tried to focus on the most important results. We think that it is very difficult to shorten the discussion, since it is necessary to describe the complex pathways and roles of the many differentially expressed genes involved.

  1. The conclusion section of the manuscript should be revised and avoid presenting a reference in conclusion.

We have now removed the reference.

Reviewer 3 Report

In the manuscript Cells-1619160 by Golchert et al, authors have aimed to identify the signaling pathways that are involved in renin-b mediated protection under ischemia-related conditions using renin-b overexpressing cardiomyoblasts (H9c2 cells). Authors overexpressed two coding DNA sequences of renin-b (1a-9 and 1-9) in H9c2 cells and performed transcriptomic profiling. Transcriptomic analysis showed that renin-b overexpression affects expression of several genes and miRNAs that are involved in Warburg-like phenotype. Authors further examined the glucose uptake and lactate consumption in renin-b overexpressing cells and observed the metabolic alteration similar to the Warburg effect upon renin-b overexpression. Authors further focused on the phosphoinositide 3-kinase (PI3K) signaling pathway as it is one the pathways that are involved in shifting oxidative phosphorylation to aerobic glycolysis. Moreover, the transcriptomic analysis showed ~109 differentially expressed genes related to PI3K/AKT and mTOR signaling in H9c2 cells upon renin-b overexpression. Authors performed western blot analysis for AKT and mTOR and their phosphorylated forms as well. They observed increase in AKT phosphorylation and no alteration in the mTOR and phosphor-mTOR level in H9c2 cells upon renin-b overexpression. Owing the association of redox signaling and Warburg effect, authors also examined ROS levels in renin-b overexpressing H9c2 cells. They observed an increase in accumulation of mitochondrial superoxide and cytosolic H2O2 in renin-b overexpressing H9c2 cells and interpretated that the cellular redox status i.e., balance between generation of ROS species and their quenching shifts towards an increased ROS generation. Taken together, in this study authors suggest that renin-b overexpression upregulates aerobic glycolysis/Warburg effect, which is similar response to metabolic remodeling under starvation conditions.

Given the paucity of the mechanism of renin-b actions, this study appears interesting and attempts to identify the pathways linking renin-b with its protective effects. Furthermore, there is an extensive data set that could further prompt hypotheses for the connection between renin-b, Warburg effect/aerobic glycolysis and cell survival. Overall, the manuscript is properly written and extensively supported by the published. Results are explained explanatory. However, considering the experimental approach and evidence provided to strengthen the conclusion, following are the concerns/points to addressed.

  1. This study is solely based on overexpression model, which is a major drawback. Examining some readouts by silencing/knocking down renin-b expression would be of good support to the conclusion.

  1. Authors have used the overexpression of ren(1a-9) and (2-9) throughout the study. What is the difference between renin-b(1a-9) and (2-9)?

  1. Abstract can be revised for background and gap/lacunae.

  1. Line 30. “our data demonstrate a renin-b induced metabolic remodeling”. Renin-b overexpression in place of renin-b.

  1. Please mention 5’ and 3’ in the primer sequences.

  1. Line 187-189. “By RT-qPCR, overexpression of Ren (1a-9) and Ren (2-9) transcript levels were measured resulting in a 2.06- and 1.74-fold overexpression of Ren(1a-9) and Ren (2-9), respectively.” 2.06- and 1.74-fold increase in expression. Is it a good overexpression efficiency? Was it statistically significant?

  1. Figure 6. Please provide representative blot images along with the quantitative graphs.

  1. Figure 7. Panel A, B and C. Does the Y-axis represent fluorescence intensity or the number of cells?

  1. Line 486-487. “Therefore, in this study we aimed to define and characterize ‘intermediate phenotypes’ linking renin-b with its effects.” What do author imply by “Intermediate phenotypes”? Please explain

  1. Please consider highlighting some implication of this studies in the discussion section.

  1. Please proofread the manuscript. Some sentences need to be rephrased in order to have clear message.

Author Response

Response to reviewer 3:

  1. This study is solely based on overexpression model, which is a major drawback. Examining some readouts by silencing/knocking down renin-b expression would be of good support to the conclusion.

We agree and thank the reviewer for the critique. However, we cannot perform such an experiment due to time constraints. Therefore, we have now included this argument in the discussion section on page 21 in lines 730-733.

  1. Authors have used the overexpression of ren(1a-9) and (2-9) throughout the study. What is the difference between renin-b(1a-9) and (2-9)?

We have now included a figure explaining the difference between both renin-b transcripts more clearly (please see the new figure 1 on page 3).

  1. Abstract can be revised for background and gap/lacunae.

Unfortunately, we are limited to a maximum of 200 words in the abstract and therefore unable to give more background information at this position.

  1. Line 30. “our data demonstrate a renin-b induced metabolic remodeling”. Renin-b overexpression in place of renin-b.

We have thankfully adopted this correction.

  1. Please mention 5’ and 3’ in the primer sequences.

We now added the 5’ to 3’ direction of the primer sequences to table 1 on page 4.

  1. Line 187-189. “By RT-qPCR, overexpression of Ren (1a-9) and Ren (2-9) transcript levels were measured resulting in a 2.06- and 1.74-fold overexpression of Ren(1a-9) and Ren (2-9), respectively.” 2.06- and 1.74-fold increase in expression. Is it a good overexpression efficiency? Was it statistically significant?

We aimed to keep the degree of overexpression rather low to avoid artifacts. Indeed, the degree of overexpression is similar to the increase in endogenous renin-b expression by glucose depletion (see ref. 5,13,14). We also performed statistical analysis of Ren(1a-9) and Ren(2-9) transcript levels in the respective overexpressing cell lines. We detected statistical significant overexpression in Ren(2-9) cells only (p=0.0403 (t-test)). This might be due to the small sample size, since we only included 3 or 4 replicates per group, respectively, that were actually also included in the transcriptome analysis. Moreover, data on renin transcript levels of the cells used in the present study to investigate AKT and mTOR protein levels and phosphorylation states with a sample size of n=6 clearly demonstrated significant overexpression of both renin-b transcripts: Ren(1a-9) and Ren(2-9) cells showed statistically significant overexpression with p=0.0327 and p=0.0172 (t-test), respectively.

  1. Figure 6. Please provide representative blot images along with the quantitative graphs.

We now provide a supplementary figure S1 showing representative western blot images for the data shown in figure 6 (now figure 7). Initially, we investigated more groups in this study than those discussed here. However, since we finally restricted our transcriptome analyses to the pIRES control, Ren(1a-9) and Ren(2-9) overexpressing groups, we also illustrated these groups in figure 6 (now 7) only. Therefore, the representative blot images also show two additional groups and we decided to include them in the supplement.

  1. Figure 7. Panel A, B and C. Does the Y-axis represent fluorescence intensity or the number of cells?

The y-axis represents the fluorescence intensity of MitoSOX+, DHE+, or DHR+ cells given in log units, respectively. We have now added an “of” in the respective y-axis legends to make it less ambiguous.

  1. Line 486-487. “Therefore, in this study we aimed to define and characterize ‘intermediate phenotypes’ linking renin-b with its effects.” What do author imply by “Intermediate phenotypes”? Please explain

Thank you for this question. By “intermediate phenotypes” we aimed to define renin‑b effects that lie between a strong overexpression and its physiological function. We have now rephrased this passage for better understanding on page 17 in lines 507-508.

  1. Please consider highlighting some implication of this studies in the discussion section.

Thank you for this suggestion. We have included an additional paragraph in the discussion on page 21 in lines 719-729.

  1. Please proofread the manuscript. Some sentences need to be rephrased in order to have clear message.

Thank you for this remark. We have tried our best and one of our co-authors who proofread the manuscript lived in England herself for a long time. Therefore, we can only offer to engage a commercial company for proofreading.

Round 2

Reviewer 2 Report

The authors have improved the manuscript.

Reviewer 3 Report

Authors have satisfactorily revised/modified the manuscript. 

A few minor things can be resolved before its publication.

1. Please be consistent about using p values. if p values are being shown in the graph panel (which is more informative way), please provide the p value in all the graph panel even if those are non-statistically significant.

2. In the statistics heading under Materials and methods section, authors say that significance level was ascribed as p<0.05.  However, the  figure 7c has a p value 0.00528. Should it be considered as significant? or else there is some typo error. Please revise for the correctness.